# Hemodynamic and Metabolic Responses to Moderate and Vigorous Cycle Ergometry in Men Who Have Had Transtibial Amputation

**DOI:** 10.3390/ijerph21040450

**Published:** 2024-04-06

**Authors:** Kionte K. Storey, Adam Geschwindt, Todd A. Astorino

**Affiliations:** Department of Kinesiology, California State University, San Marcos, CA 92096-0001, USA; store001@csusm.edu (K.K.S.); gesch001@csusm.edu (A.G.)

**Keywords:** lower-limb amputation, maximal oxygen uptake, cardiac output, ventilatory threshold, cycling, blood lactate concentration

## Abstract

Adults who have had an amputation face barriers to having an active lifestyle which attenuates cardiorespiratory fitness. Prior studies in amputees typically involve treadmill walking or arm ergometry, yet physiological responses to bilateral leg cycling are less understood. This study assessed the hemodynamic and metabolic responses to moderate and vigorous cycle ergometry in men who have had a transtibial amputation (TTA). Five men who had had a unilateral TTA (age = 39 ± 15 yr) and six controls (CONs) without an amputation (age = 31 ± 11 yr) performed two 20 min bouts of cycling differing in intensity. Cardiac output (CO), stroke volume (SV), and oxygen consumption (VO_2_) were measured during moderate intensity continuous exercise (MICE) and high intensity interval exercise (HIIE) using thoracic impedance and indirect calorimetry. In response to MICE and HIIE, the HR and VO_2_ levels were similar (*p* > 0.05) between groups. Stroke volume and CO were higher (*p* < 0.05) in the CONs, which was attributed to their higher body mass. In men with TTAs, HIIE elicited a peak HR = 88%HRmax and substantial blood lactate accumulation, representing vigorous exercise intensity. No adverse events were exhibited in the men with TTAs. The men with TTAs show similar responses to MICE and HIIE versus the CONs.

## 1. Introduction

More than two million Americans have experienced limb loss, and 28 million face an increased risk of having an amputation surgery [1]. Adults who have had an amputation complete minimal physical activity (PA) [2] and have challenges engaging in PA [3]. This inactive lifestyle may attenuate their muscle mass and fitness level which reduces their health status [4]. It is evident that the maximal oxygen uptake (VO_2_max) is lower in adults who have had an amputation versus non-injured controls (CONs). Chin et al. [5] showed a significantly lower VO_2_max and maximal workload (Wmax) in amputees versus CONs, which is important as exercise capacity is related to the activities of daily living and the ability to walk with a prosthetic in amputees [6].

Prior studies explored effects of PA on changes in aerobic fitness in adults who have had an amputation. Thirty sessions of one-leg cycling at an intensity equivalent to the ventilatory threshold (VT) significantly increased the VO_2_max and Wmax [5]. Results from a study by Chin et al. [7] revealed 36 and 26% increases in these outcomes after 6 wk of training, which were significantly different versus a non-exercise control group. These adaptations were attendant with high values of a preferred walking speed, suggesting that endurance training improves ambulation.

Studies in amputees have primarily employed one-leg cycling [5,7], upper-body cycling [8], or walking [6,9] as the exercise modality. However, little is known about responses to bilateral leg cycling, which has a larger working muscle mass which increases calorie expenditure and trains the musculature above the prosthetic leg. Cycling is accessible in fitness centers and can be performed at home. Kurdibaylo [10] required adults with a lower-limb amputation to perform graded exercise on an upper-body ergometer, and results showed attenuated cardiac function versus CONs. Exercise capacity is related to the cardiovascular system’s ability to transport O_2_, which encompasses the hemodynamic responses typically represented by changes in heart rate (HR), stroke volume (SV), and cardiac output (CO). Therefore, a diminished capacity to deliver O_2_ may reduce daily function and in turn, the health status of amputees.

The American College of Sports Medicine recommends 150 min/week of moderate intensity continuous exercise (MICE) including walking, running, cycling, or swimming, to increase fitness and health [11]. This prescription also applies to individuals with disabilities, including amputees. High intensity interval exercise (HIIE), repeated brief (5 s–5 min) efforts completed at workloads at or near VO_2_max, leads to significant increases in VO_2_max [12], glycemic control [13], and reductions in body fat [14] which are similar or superior to MICE [15,16]. To our knowledge, no study has examined the acute physiological response to HIIE of adults who have had an amputation or compared these responses to a bout of MICE.

The aim of the current study was to examine hemodynamic and metabolic responses to MICE and HIIE in men who have had a transtibial amputation (TTA) and compare these responses to those of non-amputees. Transtibial amputation consists of removing the foot, ankle joint, distal tibia, fibula, and corresponding soft tissue. Heart rate, SV, CO, VO_2_, and blood lactate concentration were measured during moderate and vigorous exercise to assess the whole-body cardiometabolic response. It was hypothesized that similar responses will be exhibited between groups. Resultant data may be applicable to clinicians who design exercise training programs for adults who have had an amputation.

## 2. Materials and Methods

### 2.1. Study Design

This study had a within-subjects crossover design requiring three sessions held at the same time of day (09:00–12:00) within participants. Sessions were preceded by no exercise in the last 24 h and the completion of a 3 h fast, which were confirmed with a written log. VO_2_max and VT were initially determined, and subsequent sessions required MICE or HIIE, the order of which was randomized across participants using a Latin squares design. Sessions were completed at least 2–7 d apart. VO_2_, hemodynamic responses, and blood lactate concentration (BLa) were measured. We used the CONSORT Reporting Guidelines in the study [17], and the trial is registered with the Open Science Framework (https://doi.org/10.17605/OSF.IO/PXGWH, accessed on 20 March 2023).

### 2.2. Participants

Five men who have had a unilateral TTA (age and time since amputation = 39 ± 15 yr and 8 ± 5 yr) and six CONs were recruited via word-of-mouth. All were healthy non-smokers without orthopedic issues. The men who have had a TTA used their prosthesis daily, and amputations were due to trauma (n = 4) or infection (n = 1). Their physical characteristics are demonstrated in Table 1, showing no significant difference (*p* > 0.05) in outcomes between groups for age, mass, BMI, %body fat, or physical activity. Written informed consent was obtained, and the study protocol was approved by the CSU—San Marcos Institutional Review Board.

### 2.3. Assessment of VO_2_max and VT

Height and body mass were determined using a balance beam scale and wall-mounted stadiometer and were used to calculate the body mass index. Skinfolds were measured in rotational order at the chest, abdomen, and thigh [18] using a Lange metal skinfold caliper to assess body fat. Then, participants completed ramp exercise on an electrically braked cycle ergometer (Velotron RacerMate, Quark, Spearfish, SD, USA). Workload began at 40–70 W for the first 2 min followed by 20–35 W/min increases until volitional fatigue occurred, identified as a pedal cadence < 50 rev/min. Throughout exercise, gas exchange data (VO_2_, VCO_2_, V_E_, and RER) were acquired every 15 s using a metabolic cart (ParvoMedics True One, Sandy, UT, USA), which was calibrated pre-testing. VO_2_max represented the mean of the two highest 15 s values at exercise termination. To verify VO_2_max attainment, these criteria were used: ΔVO_2_ ≤ 0.15 L/min at VO_2_max; HRmax ≤ 10 beats/min of 220—age, and RER ≥ 1.10 [19]. In men who have had a TTA, no adjustments were needed to the cycle ergometer to allow bilateral cycling.

The ventilatory threshold was identified as the power output showing a nonlinear increase in V_E_/VO_2_ with no commensurate increase in V_E_/VCO_2_ [20]. Before exercise, during the warm-up, and every minute during incremental testing, participants provided ratings of perceived exertion (RPE) and affective valence (FS) as described below.

### 2.4. Assessment of Thoracic Impedance

An impedance cardiograph device (PhysioFlow Enduro, Manatec, Strasbourg, France) was used to estimate hemodynamic responses [21,22,23]. Pre-exercise, participants sat quietly for 5 min, and blood pressure (Omron Tru-Gage Cuff, Omron Healthcare, Vernon Hills, IL, USA) was determined twice at the brachial artery with a 1 min period between measurements. Six electrodes (PhysioFlow Versa Trode, Nissha Medical Technologies, Devon, UK) were placed on each participant, one on the right chest, one at V5, two on the left neck, and two left of the spine at the height of the xiphoid process, according to manufacturer recommendations [21,22]. A 30-beat calibration procedure ensued to determine resting HR, SV, and CO, and participants rested for 1 min before the initiation of exercise. During exercise, outcomes were averaged every 15 s.

### 2.5. MICE and HIIE Sessions

Initially, participants rested for 5 min and 0.7 µL of blood was sampled from a fingertip using a lancet (Owen Mumford, Marietta, GA, USA) and monitor (Lactate Plus, Nova Biomedical, Waltham, MA, USA) to assess their BLa. The hand was cleaned with water and then dried with a towel. The first drop of blood was wiped away and the second drop was used. Subsequently, a 3 min warm-up began at 20%Wmax. MICE consisted of 10 min at a workload = 40% below VT and 10 min at a workload = 20% below VT. To prescribe intensities for each exercise bout, we used the VT rather than %HR/VO_2_max, as these approaches lead to inhomogeneous metabolic strain across participants. HIIE required ten 1 min efforts at 20% above the VT separated by 1 min of recovery at 20%Wmax. Moderate intensity continuous exercise and HIIE were selected as these modalities improve fitness and health-related outcomes in adults who have had an amputation as well as non-injured adults [7,12,14]. Gas exchange data and thoracic impedance were continuously acquired with values reported every 15 s, and BLa was assessed 10 min into exercise and 3 min post-exercise.

Hemodynamic variables and VO_2_ were reported as mean and peak values [24]. Peak values were determined as the mean of the three highest consecutive 15 s values at any point of exercise. Mean values included data from the entire 20 min session, excluding pre-exercise and warmup.

### 2.6. Assessment of RPE and FS

RPE (6–20) [25] and the FS [26] (11-point scale, from +5 very good to −5 very bad) were recorded pre-exercise and during MICE and HIIE. Pre-trial, participants were read instructions according to what each measure encompassed [25,26]. During HIIE, these were measured at the cessation of each interval in order of the RPE and FS. They were asked to respond to each scale in terms of their perception at that moment, and their score was repeated to them by the investigators to verify the value. The meaning of the Borg 6–20 RPE scale [10] was communicated by instructing participants to report their exertion based on their level of fatigue, breathing, and HR. This scale has anchors at values equal to 9, 11, 13, 15, 17, etc., designating “very light”, “light”, “somewhat hard”, “hard”, and “very hard”. To describe affective valence [26], we read the participants the following script: *While participating in exercise, it is common to experience changes in mood. Some individuals find exercise pleasurable; whereas, others find it to be unpleasant. Additionally, feeling may fluctuate across time. That is, one might feel good and bad a number of times during exercise.* Post-exercise, participants rested for 5 min and completed the Physical Activity Enjoyment Scale (PACES) to assess level of enjoyment using responses to 18 items on a 1–7 scale [27].

### 2.7. Statistical Analysis

Data are expressed as mean ± SD and analyzed using SPSS Version 28 (IBM, Armonk, NY, USA). We determined the normality of data distributions using the Shapiro–Wilks test, and all variables were shown to be normally distributed (*p* > 0.05). An independent *t*-test was used to compare demographic data and maximal exercise responses between groups. A repeated measures ANOVA was used to compare dependent variables including the mean and peak outcomes between groups. A three-way repeated measures ANOVA was used to compare BLa and perceptual responses, with factors including time, bout, and group. If a significant F ratio occurred, Tukey’s post hoc test was used to identify differences between means. Cohen’s d was used to estimate effect size. Statistical significance was equal to *p* < 0.05.

## 3. Results

### 3.1. Maximal Data in TTAs and CONs

Table 2 shows no difference in any outcome between groups (*p* > 0.09). Higher absolute VO_2_max, Wmax, and COmax values occurred in CONs, yet these differences were not significant (*p* = 0.09–0.23). The relative VO_2_max for men who have had a TTA was higher than that exhibited in prior studies [5,6,7] and similar to values for non-amputees [28].

### 3.2. Changes in Hemodynamic and Cardiometabolic Responses with MICE and HIIE

Table 3 compares the differences in these variables during MICE and HIIE. Other than the mean/peak SV and peak CO, there was no timeXgroup interaction (*p* > 0.08) for any variable. Post hoc analyses showed significantly higher (*p* < 0.05) peak CO (d = 2.8) and mean/peak SV (d = 2.7 and 2.7) for CONs versus TTAs. Mean (d = 1.2 and 1.3) and peak HR (d = 0.86 and 1.59), peak SV (d = 1.12), mean (d = 0.88–1.4) and peak CO (d = 1.1 and 2.2), EE (1.79 and 0.73), and mean (d = 1.33 and 1.03) and peak VO_2_ (d = 1.04 and 0.95) were significantly lower (*p* < 0.05) in response to MICE versus HIIE in both groups. Figure 1 and Figure 2 show representative exercise data for a male who has had a TTA (age = 35 yr) exhibiting an attenuated HR, CO, and VO_2_ with MICE versus HIIE, and a near-maximal SV maintained during exercise.

Figure 3 reveals the change in BLa in response to both exercise protocols in our participants; data were combined as there was no effect of group (*p* = 0.68) nor any group interactions (*p* > 0.71). The BLa significantly increased across time (*p* < 0.001) and there was a significant timeXbout interaction (*p* < 0.001), as BLa was higher in response to HIIE versus MICE at 10 min (d = 3.7) and post-exercise (d = 3.4).

### 3.3. Changes in Perceptual Responses to MICE and HIIE

The RPE significantly increased across time (*p* < 0.001) and there was a significant effect of bout (*p* = 0.013) and a timeXbout interaction (*p* = 0.005) (Figure 4). Results showed no group effect (*p* = 0.87) or any interaction (*p* > 0.48). There was a seven- and nine-unit increase in RPE (d = 3.7 and 4.7) from pre- to end-exercise during MICE and HIIE, and all RPE values were significantly different between MICE and HIIE (d = 0.77–0.91) other than those at 5 min. Results showed a significant effect of time on FS (*p* < 0.001), although there was no effect of bout (*p* = 0.17) or group (*p* = 0.22) and no other interactions (*p* > 0.12). Post hoc analyses revealed significant declines in FS values from pre-exercise to 10 min for MICE (d = 0.63) and HIIE (d = 1.2) and from 10 to 20 min in MICE (d = 1.2) and HIIE (d = 0.7). Affective valence declined by 2.0 and 3.5 units in the men who have had a TTA and CONs in response to MICE, and by 2.2 and 4.2 units during HIIE. Data showed similar PACES between MICE and HIIE (93 ± 15 and 86 ± 11 vs. 100 ± 13 and 97 ± 13 for the men who have had a TTA and CONs, *p* = 0.067) and no boutXgroup interaction (*p* = 0.72).

## 4. Discussion

This novel study compared hemodynamic and cardiometabolic responses between MICE and HIIE in men who have had a TTA versus non-injured CONs. Results showed higher HR, CO, and VO_2_ values during HIIE versus MICE, supporting data from non-amputees [24]. There was no difference in HR or VO_2_ during exercise between groups, suggesting that the two-leg cycling performed by men who have had a TTA elicits similar cardiovascular responses versus non-amputees. Lastly, men reported “good” affective valence and high enjoyment, suggesting that cycling-based MICE and HIIE are not viewed as unpleasant for men who have had a TTA.

Our study has several unique methodological elements versus prior studies in amputees. First, we prescribed exercise using a metabolic outcome (VT) rather than the %HR/VO_2_max, which reflects the cardiovascular demands of exercise. Exercise prescription using %HR/VO_2_max places participants at different levels of metabolic strain, exhibited by discrepancies in BLa which reduce exercise tolerance and elicit premature fatigue [29]. Second, we examined exercise responses below (MICE) and above the VT (HIIE) to better characterize the acute hemodynamic and metabolic response to varied exercise intensities, which can be used by clinicians for proper exercise prescription. Third, we selected two-leg cycling as it activates a large amount of muscle mass and is accessible in gyms or at home.

HIIE is unique in is its ability to elicit a near-maximal HR which is important to substantially increase VO_2_max long term [30]. The peak and mean HR in response to HIIE were equal to 88 and 80%HRmax in men who have had a TTA and CONs. Similar values were reported in non-amputees [31] and adults with spinal cord injuries [32] (SCIs) and reflect the substantial cardiovascular stimulus imposed by HIIE. Our results show no effect of group on HR/VO_2_ responses to MICE and HIIE, which opposes those of prior work in amputees. Jarvis et al. [33] reported higher VO_2_ values (+24%) in bilateral transfemoral amputees versus CONs during 5 min of self-selected walking; however, there were no differences in those who have had unilateral transtibial or transfemoral amputations. A meta-analysis [34] showed no differences in VO_2_ between CONs and adults who have had a TTA. Nevertheless, significantly higher VO_2_ values occurred in adults who have had a lower limb amputation completing walking compared to CONs [35]. Yet, other results [36] showed similar VO_2_max values between adults who have had a traumatic amputation and CONs, suggesting that the origin and region of an amputation exert significant effects on the VO_2_ response to exercise. Our data, albeit speculative due to the small and homogeneous sample, suggest that men who have had a TTA attain a near-maximal HR during HIIE, similar in magnitude to that observed in CONs, which could potentiate increases in VO_2_max values if performed long-term.

Table 3 shows significantly higher CO values in response to HIIE versus MICE, as revealed in non-amputees completing cycling [24]. CO values increased approximately 4-fold in response to MICE and 5-fold during HIIE. In addition, HIIE induced higher CO values versus that of MICE, which was explained by higher SV values. Interval exercise elicited near-maximal SV values, suggesting that vigorous exercise elicits substantial central oxygen delivery in amputees. In addition, HIIE elicited an 86 and 89%COmax in men who have had a TTA and CONs. Kurdibaylo [10] measured changes in cardiovascular outcomes in response to wheelchair ergometry up to 75% of the predicted VO_2_max in amputees. Compared to CONs, amputees showed lower SV values and higher HR values and attenuated end diastolic and end systolic volumes. In amputees, SV declined at the highest intensity, whereas, it continued to increase in CONs, suggesting superior hemodynamic function. Yet, the VO_2_max was not determined, so it is unknown if the fitness level differed across groups, which could partially explain the discrepancies reported in hemodynamic function. Vella and Robergs [37] stated that SV plateaus in untrained adults at 50–60% VO_2_max, an intensity similar to MICE, which may explain the lack of difference in the SV between MICE and HIIE shown by men who have had a TTA despite the differences in intensity between sessions. However, CONs showed higher peak SV and CO values versus those who had a TTA, which may be due to a higher body mass (+11 kg) which is attendant with a higher BV and SV. Recent data show a causal impact of BMI on cardiovascular function [38], as participants with a higher BMI reveal a higher CO, supporting this potential explanation. Unfortunately, our study cannot identify the origin of this discrepancy in SV between groups, so further testing is merited to compare hemodynamic responses to exercise in amputees versus age-, mass-, and fitness-matched controls.

Blood lactate concentration reflects the balance between lactate production through glycolysis and its removal [39]. Compared to MICE, vigorous exercise including HIIE significantly increases BLa due to a greater activation of fast twitch fibers [39]. Figure 3 revealed that BLa peaked at 8.5 mM after HIIE, supporting values from non-amputees [31] and adults with SCIs [32] completing HIIE on the cycle or arm ergometer. A peak BLa equal to 6 mM was shown in amputee soccer players [40]. Nevertheless, the peak BLa is equal to 14 mM in able-bodied elite soccer players [41], with this difference being due to the dramatically lower peak velocities attained in amputee compared to conventional soccer. Our data show no difference in BLa in response to MICE and HIIE between groups, which may be related to their similar baseline VO_2_max and VT values. Studies in non-amputees exhibit that BLa serves as an energy sensor [42], so we recommend measuring the BLa of amputees to better portray their metabolic response to acute or chronic exercise.

The perceptual response to exercise in amputees is poorly understood, which is unfortunate considering the relationship between changes in outcomes including FS and long-term exercise adherence in non-injured adults [43]. It is likely that no form of physical activity, irrespective of its efficacy, is feasible for adults unless it is well-tolerated. Data exhibit similar changes in RPE and FS values, as there was no main effect of group nor a groupXtime interaction. These results may be related to similar VO_2_max, PA, and BLa values between groups. HIIE revealed lower FS values versus those of MICE, which is related to the higher BLa attendant with exercise above the VT [31]. However, this aversive response was not coincident with lower enjoyment, as our results show no differences in enjoyment between sessions. Adults cite a lack of enjoyment as a primary barrier to regular PA [44], which necessitates designing exercise regimens which not only promote health and fitness-related benefits, but also a positive perceptual response.

This study has a few limitations. Our data do not apply to men who have had upper-body or bilateral amputations, and our sample had a VO_2_max similar to that of non-amputees, so results apply primarily to habitually active amputees. Infinite permutations of MICE and HIIE exist, and different responses would occur if the duration, intensity, or modality were modified. Our participants comprised a small convenience sample, so additional testing is needed to verify these results in a larger and more diverse group of adults who have had an amputation. Nevertheless, our results are strengthened by matching participants by age, physical activity, and VO_2_max values, so discrepancies in physical characteristics may not affect our results. Also, prescribing exercise according to a VT standardizes the acute metabolic response, which better characterizes whole-body responses to MICE and HIIE versus the use of %HR/VO_2_max [29]. Lastly, this is the first study documenting the cardiometabolic response to HIIE of men who have had an amputation, and our results may apply to clinicians and scientists who implement physical activities for this population.

## 5. Conclusions

Results show no significant differences in HR, VO_2_, hemodynamic, or perceptual responses to MICE and HIIE in men who have had an TTA versus controls. HIIE elicited near-maximal HR and BLa values, demonstrating that this mode elicits vigorous responses in amputees similar to those seen in non-injured adults. No adverse events were reported during HIIE and participants who had a TTA rated the bout as enjoyable. These results, albeit preliminary, support the utility and feasibility of bilateral HIIE cycling for healthy, active men who have had a unilateral lower-limb amputation.

## Figures and Tables

**Figure 1 ijerph-21-00450-f001:**
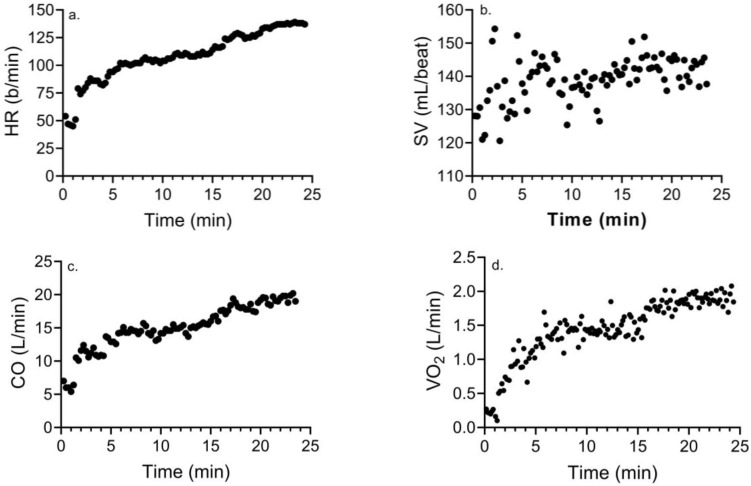
Changes in (**a**) HR, (**b**) SV, (**c**) CO, and (**d**) VO_2_ in response to MICE in a male who has had a TTA.

**Figure 2 ijerph-21-00450-f002:**
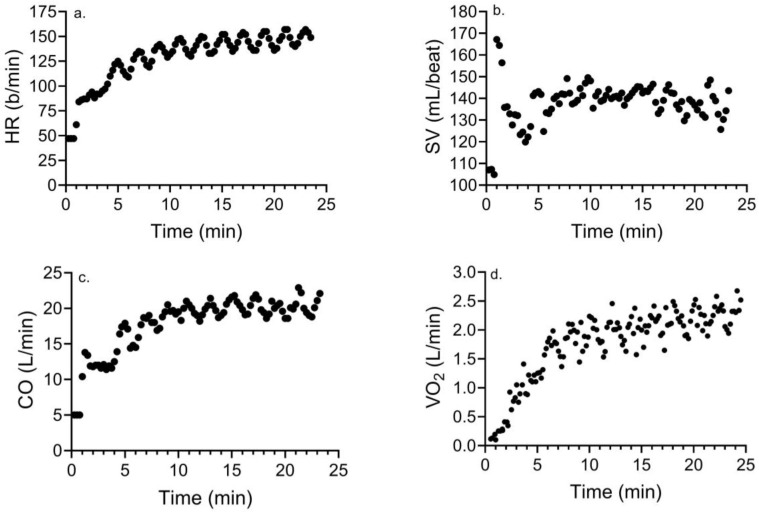
Changes in (**a**) HR, (**b**) SV, (**c**) CO, and (**d**) VO_2_ in response to HIIE in a male who has had a TTA.

**Figure 3 ijerph-21-00450-f003:**
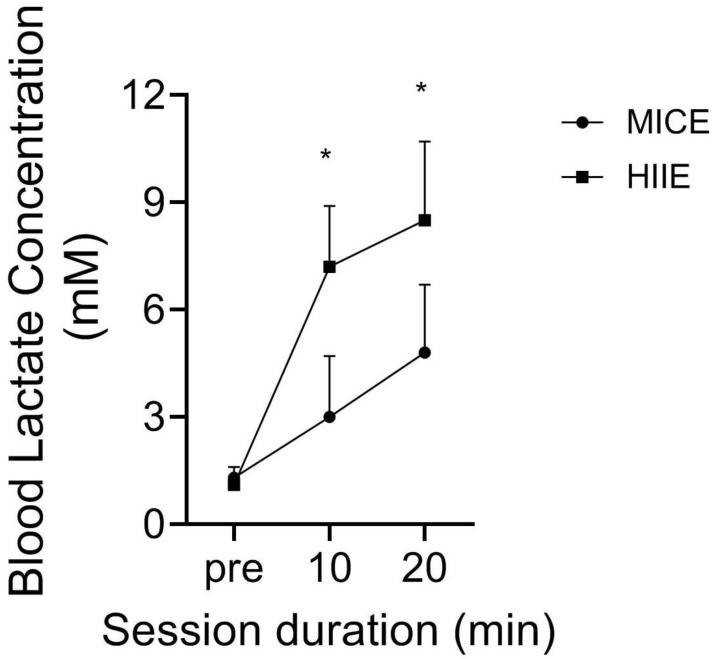
Change in blood lactate concentration in response to MICE and HIIE (mean ± SD). * *p* < 0.05 between MICE and HIIE.

**Figure 4 ijerph-21-00450-f004:**
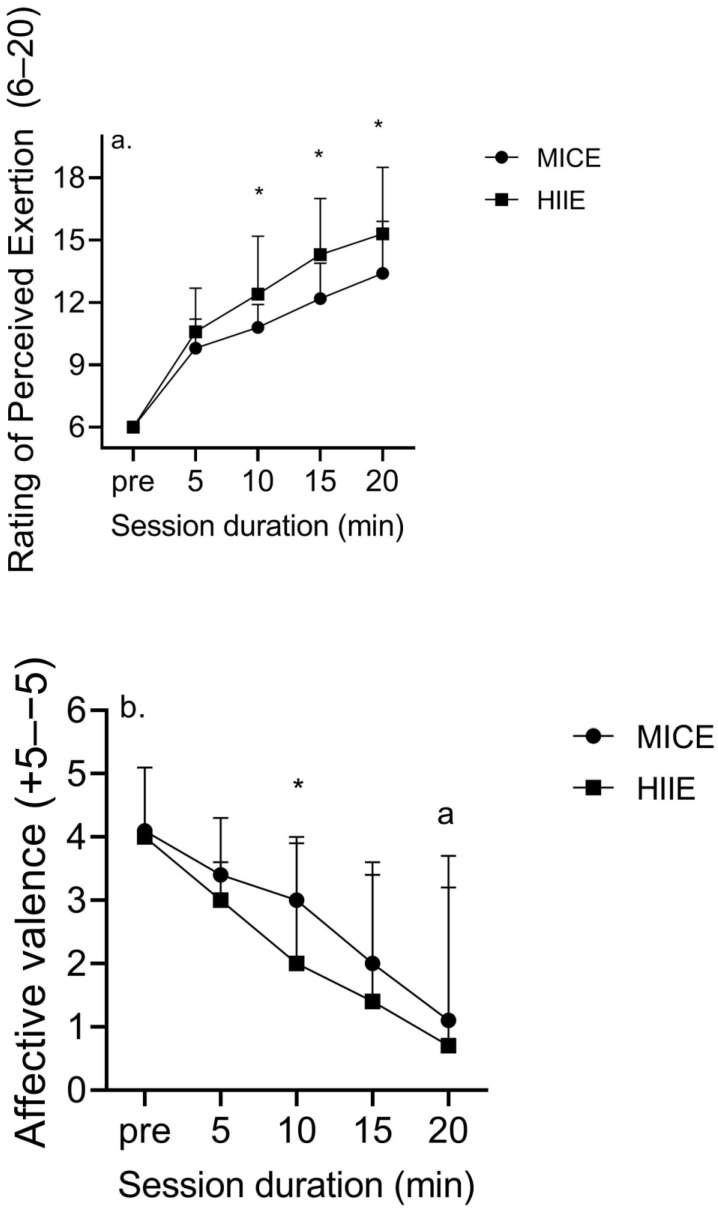
Changes in the (**a**) RPE and (**b**) affective valence in response to MICE and HIIE in men who have had a TTA and CONs (mean ± SD). Data were combined due to there being no difference in either outcome between groups. * *p* < 0.05 between MICE and HIIE (**a**); * *p* < 0.05 versus pre-exercise value (**b**); ^a^
*p* < 0.05 vs. 10 min value.

**Table 1 ijerph-21-00450-t001:** Comparison of demographic traits between men who have had a TTA (n = 5) and CONs (n = 6) (mean ± SD).

Outcome	TTAs	Range	CONs	Range	*p* Value
Age (yr)	39 ± 15	21–61	32 ± 11	21–50	0.37
TSA (yr)	8 ± 5	0.5–12.0	NA		
Body mass (kg)	75 ± 10	60–86	86 ± 11	69–99	0.11
BMI (kg/m^2^)	23.8 ± 3.5	20.3–29.1	27.0 ± 3.4	23.0–33.3	0.08
Body fat (%)	14.4 ± 6.9	7.1–26.1	18.3 ± 6.1	15.3–25.2	0.37
Physical activity (h/wk)	7.8 ± 4.5	1–12	5.3 ± 1.4	3–7	0.22

TTAs = men who have had a transtibial amputation; CONs = controls; TSA = time since amputation; NA = non applicable; BMI = body mass index.

**Table 2 ijerph-21-00450-t002:** Maximal responses to incremental cycle ergometry in men who have had a TTA (n = 5) and CONs (n = 6) (mean ± SD).

Outcome	TTAs	Range	CONs	Range	*p* Value
Wmax (W)	256 ± 44	191–302	298 ± 48	251–359	0.16
VO_2_max (mL/kg/min)	39.2 ± 7.5	28–46	40.2 ± 3.6	35–46	0.76
VO_2_max (L/min)	2.89 ± 0.42	2.38–3.49	3.43 ± 0.50	2.80–4.05	0.09
V_E_max (L/min)	151.1 ± 9.5	137–160	147.7 ± 20.5	115–218	0.85
RER	1.30 ± 0.05	1.24–1.37	1.24 ± 0.10	1.12–1.36	0.27
HRmax (b/min)	181 ± 11	169–193	182 ± 9	166–191	0.89
SVmax (mL/beat)	118 ± 25	82–142	134 ± 24	117–180	0.30
COmax (L/min)	22.0 ± 4.1	15–26	23.8 ± 2.7	20–28	0.23
BLa (mM)	11.1 ± 1.5	8.9–12.6	10.7 ± 1.4	8.2–12.1	0.70
VT (%VO_2_ max)	65 ± 4	61–72	66 ± 5	55–70	0.86

TTAs = men who have had s transtibial amputation; CONs = controls; Wmax = maximal workload; VO_2_max = maximal oxygen uptake; V_E_max = maximal minute ventilation; RER = respiratory exchange ratio; HR = heart rate; SV = stroke volume; CO = cardiac output; BLa = blood lactate concentration; VT = ventilatory threshold.

**Table 3 ijerph-21-00450-t003:** Mean and peak hemodynamic and cardiometabolic responses to MICE and HIIE in men who have has a TTA and CONs (mean ± SD).

Outcome	TTAs		CONs		Bouts	BoutsXgroup
	MICE	HIIE	MICE	HIIE		
EE (kcal)	159 ± 16	183 ± 42 *	196 ± 31	216 ± 29 *	0.02	0.72
Mean HR (b/min)	130 ± 14	144 ± 12 *	126 ± 10	141 ± 12 *	<0.001	0.76
Mean SV (mL)	114 ± 17	113 ± 16	114 ± 18	127 ± 20 ^a^	0.054	0.02
Mean CO (L/min)	14.7 ± 1.3	16.1 ± 2.1 *	14.4 ± 1.8	17.4 ± 2.8 *	0.001	0.08
Mean VO_2_ (L/min)	1.47 ± 0.15	1.83 ± 0.40 *	1.84 ± 0.25	2.10 ± 0.30 *	<0.001	0.42
Peak HR (b/min)	149 ± 17	161 ± 14 *	144 ± 12	160 ± 10 *	0.002	0.55
Peak SV (mL)	118 ± 16	120 ± 13	119 ± 12	134 ± 17 *^,a^	0.006	0.02
Peak CO (L/min)	17.4 ± 1.4	19.0 ± 1.9 *	17.0 ± 1.9	21.5 ± 2.5 *^,a^	<0.001	0.007
Peak VO_2_ (L/min)	1.82 ± 0.21	2.17 ± 0.49 *	2.27 ± 0.31	2.58 ± 0.40 *	<0.001	0.79

MICE = moderate intensity continuous exercise; HIIE = high intensity interval exercise; TTA = transtibial amputation; CONs = controls; EE = energy expenditure; HR = heart rate; SV = stroke volume; CO = cardiac output; VO_2_ = oxygen consumption; *p* values are listed for the main effect of bouts as well as the boutsXgroup interaction; * = *p* < 0.05 within group versus MICE; ^a^ = *p* < 0.05 versus the value for HIIE in men who have has a TTA.

## Data Availability

All data relevant to this study are contained in this article.

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
