# Peer review of "Hemodynamic and Metabolic Responses to Moderate and Vigorous Cycle Ergometry in Men Who Have Had Transtibial Amputation"

_ijerph, 2024, doi:10.3390/ijerph21040450_

Round 1
Reviewer 1 Report
Comments and Suggestions for Authors
Author Response
We have replied to all of your concerns in the enclosed rebuttal and have made corresponding changes to the paper. We hope this satisfies all of your concerns.

Reviewer 2 Report
Comments and Suggestions for Authors
Line 36- how? greater than the ctrl group?
line 40- have employed
Line 49- define ACSM?
Line 65- space between 2.1 and study
Line 126- are these previously used protocols?
Line 138- is this overall RPE?
This is a very interesting and overall well written article regarding an understudied topic. I have few comments, listed above, for clarification purposes.
Author Response

(The authors gave the same response as above.)

Reviewer 3 Report
Comments and Suggestions for Authors
This study is meaningful because it is a study of hemodynamic and metabolic responses to moderate and vigorous cycles of ergometry in men who underwent tibia amputation, and it is a study of amputees that are not easy to recruit.
However, there are some questions as described below.
In Figures 1 and 2, the graphs a, b, c, and d should be accurately indicated.
Line 138 & Line 299-301 : Participants with TTA evaluated the match as enjoyable, as mentioned in the conclusion. Where can I see the results analysis for the Activity Enjoyment Scale?
Line 254-255 : The authors stated that the evidence of higher peak SV and CO in TTA was due to higher body mass, so why do you think so and is there a basis for that? It should be possible to provide evidence to support the claim.
Line 288-290 & Table 1 : The authors stated that age, physical activity, and VO2max match. For this argument to be accepted, a normality test should be performed in the analysis in Table 1. However, since Table 1 doesn't know this content, authors need to present the p-value for the difference between groups.
Author Response

(The authors gave the same response as above.)

Reviewer 4 Report
Comments and Suggestions for Authors
From my point of view, the work in general is correctly done, on a strong application basis, which reveals the practical authenticity of the results and the research approach carried out by the authors.
My only suggestion for proofreading the paper refers to the Introduction. The authors should add additional information regarding what hemodynamic responses represent in general (as well as conceptual delimitations), respectively a short theoretical description of what transtibial amputation means. It is essential that future readers correctly understand the concepts indicated in the title of the work.
Author Response

(The authors gave the same response as above.)

Round 2
Reviewer 1 Report
Comments and Suggestions for Authors
I appreciate the effort made. Continued success!
Author Response
We thank the Reviewer for his/her positive comments concerning our paper. No changes have been made to the paper at this point, as we assume from the comment listed below that the Reviewer is satisfied with the previous changes made to the paper and our responses to initial concerns. Thank you for reviewing our submission.
Reviewer 3 Report
Comments and Suggestions for Authors
VO2max (p=0.76), but no related figures are shown in Table 1.
It is also questionable to argue that there is no significant difference in body mass between amputated patients and normal adults. In addition, the authors do not seem to understand the basic statistical term for normality testing.
And in general, p-values are displayed in tables rather than in text, so that readers can understand easily.
In this study, the overall expression of the results is poor, and there is a lack of evidence to support the authors' arguments. In addition, communication with reviewers should be done in a point-to-point format, and it is regrettable that it is insufficient in this area.
Author Response
We appreciate your new comments concerning our paper, and below, we have uploaded a new point-by-point rebuttal which we hope satisfies these concerns; thank you. We have also uploaded a revised version of our manuscript per your comments.
